# A Methodology for Training Toolkits Implementation in Smart Labs

**DOI:** 10.3390/s23052626

**Published:** 2023-02-27

**Authors:** Majid Zamiri, Joao Sarraipa, José Ferreira, Carlos Lopes, Tal Soffer, Ricardo Jardim-Goncalves

**Affiliations:** 1School of Science and Technology and Center of Technology and Systems (CTS-Uninova), NOVA University of Lisbon, 2829-516 Caparica, Portugal; 2Unit for Technology and Society Foresight, School of Education, Tel Aviv University, Tel Aviv 69978, Israel

**Keywords:** training, toolkits, smart labs, skills development

## Abstract

Globally, educational institutes are trying to adapt modernized and effective approaches and tools to their education systems to improve the quality of their performance and achievements. However, identifying, designing, and/or developing promising mechanisms and tools that can impact class activities and the development of students’ outputs are critical success factors. Given that, the contribution of this work is to propose a methodology that can guide and usher educational institutes step by step through the implementation of a personalized package of training Toolkits in Smart Labs. In this study, the package of Toolkits refers to a set of needed tools, resources, and materials that, with integration into a Smart Lab can, on the one hand, empower teachers and instructors in designing and developing personalized training disciplines and module courses and, on the other hand, may support students (in different ways) in developing their skills. To demonstrate the applicability and usefulness of the proposed methodology, a model was first developed, representing the potential Toolkits for training and skill development. The model was then tested by instantiating a particular box that integrates some hardware to be able to connect sensors to actuators, with an eye toward implementing this system mainly in the health domain. In a real scenario, the box was used in an engineering program and its associated Smart Lab to develop students’ skills and capabilities in the areas of the Internet of Things (IoT) and Artificial Intelligence (AI). The main outcome of this work is a methodology supported by a model able to represent Smart Lab assets in order to facilitate training programs through training Toolkits.

## 1. Introduction

In the global, fast-changing educational ecosystem, modern trends and approaches to education have brought about a significant positive alteration to the education system and facilitated the process of training and learning. A modernized approach to education refers to the use of effective methods and supportive technologies, aiming to improve the quality of education. Under the influence of methodological achievements and technological advances, modern approaches to education have made training and learning more engaging, useful, and interesting [1]. Unlike the traditional and classical methods of education that are basically restrictive and textbook-based, the modern approaches to education not only emphasize sociable interactive environments but also offer a broad spectrum of training and learning materials and tools that can in turn promote lifelong learning skills [2].

Nowadays, one of the major challenges for the educational system (particularly higher education) is to bridge the gap between academia and industry toward preparing and training skillful students (the future workforce) that can truly meet the demands of the labor market [3]. Furthermore, traditional teaching approaches have not deeply focused on the close relationship with industry requirements. For example, one problem with the traditional approaches to training and learning is that the graduated students in such a system are not developed (or, in some cases, they are developed very poorly) with the needed industrial-oriented learning experience prior to their first job [4]. Despite the fact that businesses around the world continue to absorb the skilled workers they need. For the reason that skillful workforces—as a source of competitive advantage—not only enable businesses to offer their consumers greater value but also help businesses outperform their competitors. Indeed, when the average skill in a specific labor market is relatively low, the businesses have to either make some strategies to train and up-skill sufficiently their current workforce (which could be costly, time-consuming, and challenging for them) or decide to invest in hiring the trained and skilled workforce (which can save time and cut their expenses) [5]. 

From this perspective, the functions of higher education institutions are now shifting from academic-based (teaching and research) to industry-oriented (training and upskilling) in order to introduce the shortest feasible paths for students to the professional world. Industrial-oriented learning is an approach to learning from an industry perspective, meaning that it helps students be prepared for developing further knowledge and skills required for a target industry. It provides several opportunities for students that can challenge them to develop their practical, social, and intellectual skills in a holistic way, aiming to keep up with the industry and become involved in the industry. For example, ref. [6] claims that Education 4.0 refers to a broad trend in preparing the future workforce for Industry 4.0 rather than a single solution. It encompasses various ways that higher education institutions are adapting their offerings and curricula to better equip future graduates for the workplace.

Nowadays, there are lots of higher education institutions in the world that are trying to create, develop, and promote various industry-oriented course modules and job-oriented training programs. Such courses and programs can potentially have a great impact on students’ future careers and businesses’ growth. Expectantly, they are designed and adjusted (by professionals from academia and/or industry) to help students obtain both the theoretical and the practical knowledge, experience, and skill sets relevant to the industry [7]. In this direction and by considering the needs of industries and students, this study aims to propose a methodology that can guide and support the process of implementation of training Toolkits in Smart Labs built for this purpose. By using this approach, which is based on experiential learning principles, a holistic learning space is created wherein learning transactions take place between individuals and the environment [8]. 

The proposed training Toolkits emphasize the development of core industrial and technical skills in students, and the expectation is that they could effectively nurture the STEM talent and 21st-century skills needed nowadays. Additionally, the proposed training Toolkits are purposefully designed to provide the possibility for making synergy between foundational (hard) skills and (soft) 21st-century skills. The provided training Toolkits then be available to students and teachers for use in different tailored modern training programs and integrated Smart Labs. The primary objective of this study is well in line with the 4th goal of the 2030 Agenda for Sustainable Development of the United Nations. The 4th goal expects equitable quality education and promotes lifelong learning opportunities for people all over the world [9]. The matrix model shown in Figure 1 gives an overview of the main dimensions and functions we considered for achieving the goal of this work.

The matrix model (on the left side) addresses three main dimensions for the organization of the specific needs and resources for training, namely, (D1) identifying particular Toolkits for supporting training courses (this is the what dimension), (D2) identifying relevant disciplines/training courses that are demanding (this is the how dimension), and (D3) identifying the target students who are eager to attend the training courses (this is the who dimension). 

Having identified and managed these three dimensions (by the training providers and developers in a higher education institution), students can then be engaged in various personalized training activities and practices (representing the functions) through, for example, an integrated Smart Lab. The Smart Lab provides a supportive environment that enables students to, for example, use the provided training resources and materials; conduct research, investigation, and experimentation on the topics of interest; and engage in collaboration, communication, group discussion, and knowledge sharing with their peers, aiming to gain the needed skills and competences and then be prepared for their future jobs. This is a kind of modern approach and innovative training and learning service that intends to help students have an actual feel of the industry, make them compatible with industrial requirements, assist them in becoming effective and productive workforce, and help them in their placements. However, to optimize the approach and service, they should be initially adjusted and customized based on cultural and local needs. 

The rest of the paper is organized as follows. In Section 2, we propose our methodology for supporting the implementation of training Toolkits in Smart Labs which steers the structure of this paper. In Section 3, we explain the approach used for collecting and analyzing the data, review the related works briefly, and present a generic conceptual model for creating a package of training Toolkits in the Smart Labs. In Section 4, the instantiations we designed and used for supporting teachers and students are presented, and it briefly looks into possible future studies. Section 5 draws some conclusions and derives policy implications from our results. 

## 2. Materials and Methods

To gain an up-to-date overview of the methodologies used in similar studies for supporting the implementation of training Toolkits, an extensive literature review was undertaken. However, there is a lack of prior research on our topic. Moreover, there is no clear evidence in the literature that shows how training Toolkits can be scientifically and appropriately designed, implemented, and developed for the purpose of skill improvement in Smart Labs. This limitation, thus, leads to not gaining strong and sufficient information and evidence of related ideas, possible solutions, and potential opportunities and challenges. Hence, for designing our method and in order for us to be inspired by relevant methods, we referred (as an alternative) to the life cycle of the methods proposed, e.g., for prototyping [10], modeling and simulation [11], tool and product [12], technology development [13], system development [14], and software development [15]. We do so because we believe that the life cycle method can provide a kind of management approach that puts the Toolkits in the life cycle thinking basket into practice. The life cycle method can also offer an analytical and sequential procedure that clarifies the process of designing, implementing, evaluating, and developing the Toolkits. Reviewing such life cycles gave us some insight into the staged and workflow method that we proposed for supporting the implementation of training Toolkits in Smart Labs. As shown in Figure 2, the proposed workflow methodology encompasses two main cycles: A.*Terminological cycle*—includes five stages of the method (S1–S4 and S9) and refers to the process of modeling information to represent Smart Lab. This comprises steps for investigation and systematic study to discover, gather, and analyze the related and latest findings about the target topic toward proposing potential and useful solutions and services (e.g., discipline/module course, Smart Lab, and training Toolkits).B.*Instantiation and specification cycle*—includes the other stages (S5–S8) of the method and refers to the process of instantiating Smart Labs assets. This may result in assistance to teachers in training implementation and helping students develop their skills through Smart Labs.

The nine main stages of the proposed workflow methodology are explained in the following:A.In Terminological Cycle:
*Investigation (S1)*—refers to deep (systematic) research and detailed study on the main subject of a Smart Lab toward gaining an insight into what has been completed by similar works.*Identifying (S2)*—refers to obtaining needed information, finding evidence and examples of related works, and recognizing the needs of the Smart Lab’s subject.*Analysing (S3)*—refers to reviewing and examining in detail the identified information, evidence, and examples, as well as considering the needs for the proposed Smart Lab’s services and solutions.*Modelling (S4)*—refers to building modeling structures of the proposed Smart Lab’s services and solutions to allow the instantiation of them later.
B.In Instantiation and Specification Cycle:
*Instantiation (S5)*—refers to the formal representation of a created Smart Lab’s asset.*Using (S6)*—refers to applying the represented assets in the real world. Such assets may include the Smart Lab and/or the discipline/module course services. For example, in the case of discipline/module courses, it represents their use for learning purposes.*Evaluation (S7)*—refers to assessing and determining whether or not the used services and solutions are found useful and effective by users (e.g., teachers, students).*Improvement (S8)*—refers to making the needed changes to proposed services and solutions or adapting new solutions (e.g., Toolkits) to promote the effectiveness of the assets.*Maintenance (S9)*—refers to efforts taken to preserve, maintain, and improve the proposed Smart Lab model to properly represent or characterize its assets (e.g., Toolkits).


It should be noted that the proposed workflow methodology in this study simply conceptualizes the main stages, elements, and aspects that need to be taken into account in the implementation of training Toolkits for developing disciplines and/or students’ skills. The methodology gives some directions to researchers and developers who deal with this issue. However, since the process is dynamic, the number of cycles and stages can be increased or decreased, depending on the objectives, requirements, and conditions of the used case. This issue is clarified in Section 4.2 in Figure 8.

In the following Sections, we explain the way in which we used the proposed workflow methodology to steer this study.

## 3. Terminological Cycle

Section 3 presents the terminological cycle. It presents the four first stages of our proposed workflow methodology. This section first explains the research approach we used to collect the data. Furthermore, it presents briefly the background information (addressing S1, S2, and partially S3), the results of data analysis (S3), and the proposed model (S4) for creating a package of training Toolkits. 

### 3.1. Research Approach

We performed a deep literature review to identify and collect related and potential training Toolkits. With the intention of searching and choosing relevant papers, databases such as Web of Science, SCOPUS, Google Scholar, and IEEE Xplore were used. In addition to database searching, we reviewed pertinent commercial websites to ensure that we did not miss anything that might be relevant and also to find further Toolkits that were not addressed in the scientific papers. The results of that search led to the creation of a long list of Toolkits that can be used at different stages of the process of training and learning. After removing the Toolkits that had some similarities in the futures and characteristics considered, we finally came up with the 36 training Toolkits presented in Section 3.2.2. However, for analyzing and categorizing the identified training Toolkits we could not find in the literature an appropriate taxonomy and classification, which means that this study is at a preliminary stage, and we have reasons to believe it contains elements worth discovering to inform future research. We thus adopted our own approach and provided more detailed information about the identified Toolkits by analyzing them according to their main components and grouping them based on the fields of study they can be used in.

### 3.2. Background Information

Section 3.2.1 and Section 3.2.2 provide a brief overview of related work and explain succinctly the main concepts of this study (Smart Lab and Toolkit), addressing the first stage of our methodology (investigation). 

#### 3.2.1. Smar Labs (S1)

In today’s fast-changing world of technology and digital transformation in education, laboratories have a significant role to play. That is, they can provide an equipped environment to then take the opportunity to facilitate and promote the processes of, for example, providing an experimental foundation for understanding theoretical concepts, conducting practical tests, carrying out scientific research, developing innovation, advancing technologies according to current demands, and developing students’ skills, i.e., problem-solving and critical-thinking skills [16]. As the demands and expectations of industries, academia, and users are continually changing, the new generation of labs (e.g., industrial automation laboratories [17], living labs [16], virtual cloud labs [18], digital labs [19], and virtual reality labs [20]) are accordingly evolving. Smart Lab (as another example) is, indeed, a supportive environment, high-performance laboratory, and open workspace equipped with needed Toolkits and tools designed to enable and expedite safe and efficient world-class science and experimentation [21], as well as enhance effective communication and successful engagement in instruction, curriculum, and learning [22]. 

Smart Lab can “facilitate the provision of all kinds of learning, including traditional face-to-face and distance learning, and also offers possibilities for further development of mobile learning, which is rapidly spreading in all fields of education” [23]. Smart Labs, thus, can be built in the form of physical, virtual/mobile, or mixed labs that promote training and learning courses and activities in both the traditional mode (face-to-face) and the remote mode, depending on the training structure and purpose. Therefore, Smart Labs introduce and offer equipped, flexible, and dynamic spaces for users (e.g., teachers and students), intending to facilitate access to a wide variety of training and learning materials and resources; expedite the investigation, research, and experimentation process; promote collaboration and knowledge sharing; develop students’ skills, etc. [20]. 

Studies [21,22] indicate that because Smart Labs are not static environments by nature and they are continually adapting themselves, for example, to the current needs of their users, developing new collaborative cultures, driving innovation, and embedding the latest technologies in education, they can pave the way for augmented education through accelerating and enhancing learning processes. 

Evidence shows that Smart Labs can, on the one hand, empower teachers to benefit from experimental learning scenarios and advance their teaching techniques. Additionally, Smart Labs can provide teachers (through different evaluation methods and tools) with the needed information about each student’s progress, needs, and problems. Furthermore, the teachers can readily respond to each student’s specific requirements. On the other side, Smart Labs can empower students by providing useful and effective means for carrying out classroom activities and by facilitating engagement with immersive experiences. More importantly, Smart Labs can boost student-teacher interactions (e.g., by sending messages or exchanging files) and student-teacher relationships (by providing some opportunities for them to be involved in different social activities, i.e., workshops) [22,24]. Despite the numerous advantages and applications of Smart Labs to education, training, and learning (that were mentioned above in part), the attention in this study is given to its role in developing students’ skills and developing disciplines/module courses by teachers.

#### 3.2.2. Toolkits (S1)

The literature shows that the development of physical and electronic/digital Toolkits used for supporting the processes of training and learning has been growing since 1980 [25]. To date, numerous useful training Toolkits have been introduced to support, for example, specific users (teachers [26], students [27], children [28]), disciplines (machine learning [29], physical computing [30]), and/or skills (programming [25]). Research [27] declares that many training Toolkits have been designed to meet the specific needs of students and increase their productivity in inclusive settings. Additionally, training Toolkits can, for example, develop students’ practical, social, and cognitive skills [28], writing skills [31], advocacy skills, and critical thinking skills [32], English language proficiency, and academic literacy skills [33], and technical, communication, leadership, teamwork, and analytical thinking skills [34]. On the other hand, training Toolkits are valuable sources for teachers to modernize their approach to teaching. Training Toolkits can support teachers in different ways, including but not limited to enabling them to develop their own educational applications and activities [35], monitor and analyze their teaching activities [36], and design courses and enrich teaching [37]. 

A toolkit is an assembly of tailored tools and/or a special set of basic components, guidelines, templates, adaptable resources, and software utilities that provide practical solutions, advice, guidance, processes, and information for a particular purpose [38]. In the context of training and learning, a toolkit refers to a collection of original and/or revised resources, training classes, online courses, training materials, web links, instructional methods, models, techniques, publications, knowledge, tools, devices, and applications that can be used to, for example, translate theory into practice, expedite the translation of evidence into practice, facilitate practice change, give actionable learning paths, upskill trainers and trainees, support learners and learning strategies and skills, as well as improve the design, implementation, delivery, and evaluation of training courses [38,39]. There are different types of Toolkits (e.g., education toolkits, technical toolkits, design toolkits, collective action toolkits [40] official toolkits, expert toolkits, and basic toolkits [41]) and they are often created and used for different purposes (e.g., language model training [42], standardized reinforcement learning [43], and facilitating the engagement with learning outcomes [44]), which may focus on a specific group of users (e.g., higher education institutes [45] and healthcare providers [46]).

##### Identified Training Toolkits (S2 and Partially S3)

In order to identify and document a number of commercial Toolkits that are designed and used for training and learning purposes, we reviewed the related literature and several websites. A total of 36 training Toolkits are then identified and listed in the following (covering the second stage of our methodology), coupled with their main characteristics, namely, main users, main components, main features (addressing the strength of the Toolkits), and aims. Furthermore, the Toolkits are classified based on the field in which they can be used in order to be organized, identified, and presented easier (partially covering the third stage of our methodology).

It should be noted that this list is not exhaustive, as there should be a wide spectrum of available Toolkits, and new technological solutions emerge continuously, but it can give us a good overview of the current state of the art, available toolkits in the market, and trends in this realm.

Toolkits applicable in the field of machine learning:(1)Unity ML-Agents Toolkit [47]
-Main users: game developers, AI researchers, and hobbyists.-Main components: it comprises deep reinforcement learning algorithms (e.g., actor-critic, proximal policy, and deep deterministic policy gradients and their variants).-Main features: it supports multiple environment configurations and flexible training scenarios; it also supports the training single-agent, multi-agent cooperative, and multi-agent competitive scenarios.-Aims: it aims to enable games and simulations to serve as environments for training intelligent agents.-Fields of use: machine learning.
(2)OpenAI Gym Toolkit [48]
-Main users: it is open to any type of user.-Main components: it consists of a growing suite of environments (from very simple games to complex, physics-based gaming engines) written in Python and a site for comparing and reproducing results.-Main features: it supports teaching agents everything from walking to playing games; it also practically supports the evaluation and comparing the Reinforcement Learning agents in a generic way.-Aims: it aims to implement reinforcement learning in simulation environments.-Fields of use: machine learning (reinforcement learning). 
(3)TensorFlow Toolkit [49]
-Main users: (TensorFlow Lite) developers.-Main components: it is a suite of tools, techniques, tutorials, examples, and other resources for optimizing machine-learning models.-Main features: it supports the management of all aspects of a machine learning system, it also makes machine learning and developing neural networks easier and faster.-Aims: it aims to support and speed up model building and create scalable machine-learning solutions.-Fields of use: machine learning.
(4)DeepMind Control Suite [50]
-Main users: (DeepMind) researchers, developers, and engineers.-Main components: it is a set of Python libraries and control tasks (written in Python) with a standardized structure and interpretable rewards.-Main features: it brings about a similar set of standard benchmarks for continuous control problems. It also provides several well-tested and stable control tasks that can be easily used and modified.-Aims: it intends to serve as a performance benchmark for reinforcement learning agents.-Fields of use: machine learning (reinforcement learning).
(5)LEAF [51]
-Main users: developers.-Main components: learning techniques (based on Collective Intelligence), a set of agents and communities, FIPA platform agents, and API.-Main features: it provides an implementation of several learning techniques that support the development of learning, and it also supports the dynamic assignment of utility functions.-Aims: it aims to coordinate the behaviors of communities of learning agents and support the dynamic assignment of utility functions.-Fields of use: machine learning.


Toolkits applicable in the field of health & health care:(6)Training Toolkit [52]
-Main users: training coordinators, curriculum developers, and trainers.-Main components: it is a collection of resources (for developing, delivering, and evaluating training on HIV-related topics and skills for healthcare providers).-Main features: it provides multiple tools on specific training topics and for different training needs, and it supports quality training tailored to different target audiences.-Aims: it aims to support the preparation and presentation of HIV/AIDS training.-Fields of use: healthcare (HIV/AIDS training).
(7)Laboratory Quality Management System Training Toolkit [53]
-Main users: trainers, laboratory directors, quality managers, and laboratory technologists.-Main components: it consists of training sessions, modules, and guidelines.-Main features: it is designed based on internationally recognized standards, ISO CLSI GP26-A3, and it can support all stakeholders in health laboratory processes.-Aims: it is intended to provide comprehensive materials that will allow for designing and organizing training workshops.-Fields of use: health care (health laboratory).
(8)Training Toolkit for teachers and educators [54]
-Main users: teachers and educators.-Main components: resources and materials (10 online and short-term training units).-Main features: it provides self—paced training and learning, so it does not need the presence of an external teacher. It enables teachers with knowledge on how to support their students individually. It designs teaching methodologies according to students’ requirements.-Aims: it aims to increase teachers’ and educators’ knowledge of the latest findings in neuropedagogy, and support them with more accurate, up-to-date, and scientifically based training.-Fields of use: neuropedagogy.


Toolkits applicable in the field of the Internet of Things (IoT):(9)STEM educational Toolkit [55]
-Main users: researchers, engineers, and students.-Main components: it comprises an input/output test board, an analog-to-digital Converter, and Raspberry Pi 3 Model B+.-Main features: it exposes students to the basics of data acquisition and control from devices; it also helps students to perform remote monitoring, visual data analysis, and data processing over the Internet.-Aims: it aims to tailor STEM learning purposes at reasonable effort and cost and to support IoT-facilitated STEM education.-Fields of use: IoT.
(10)Real-time Distributed Toolkit [56]
-Main users: children.-Main components: it consists of cube-shaped wireless modules (input and feedback modules), sensors, an LCD display, a motor, RGB LEDs, and a speaker.-Main features: it operates in a distributed fashion to facilitate the exploring of IoT concepts for children. It helps children to understand the basics of IoT technologies by linking devices using rule-based systems and connecting the toolkit to smart home scenarios.-Aims: it aims to decrease the barrier of entry to primary school children’s exploration of IoT concepts.-Fields of use: IoT.
(11)ConnectUs [57]
-Main users: children.-Main components: interactive sensing, actuator cubes, and Bluetooth.-Main features: it encourages creative crafting and tinkering; it also enables children to design their own IoT system. ConnectUs is extensible to an unlimited variety of activities.-Aims: it aims to engage children with complex IoT concepts.-Fields of use: IoT.


Toolkits applicable in the field of Artificial Intelligence (AI):(12)MakeBlock AI & IoT Education Toolkit Add-on Pack [58]
-Main users: students.-Main components: it contains 31 mBuild’s electronic modules and 10 accessory packs-Main features: it enables students to apply the technology to everyday life, it also stimulates students’ imagination and curiosity. This Toolkit is compatible with different scenarios (e.g., computer science class, makerspace, robot competition, and robot after school club).-Aims: it aims to support students in learning AI, applying the technology to everyday life, and completing engaging projects by using sensors and visual programming.-Fields of use: AI.


Toolkits applicable in the field of computer science:(13)Talkoo Toolkits [59]
-Main users: teachers, students, and children.-Main components: physical computing plug-and-play modules, visual programming, and prototyping material. -Main features: it makes effective collaborative learning and physical computing for young users; it also develops students’ practical, social, and cognitive skills by doing.-Aims: it aims to improve students’ motivation and collaboration skills in project-based physical computing activities.-Fields of use: computer science (physical computing).
(14)Training Toolkit [60]
-Main users: youth workers and educators.-Main components: detailed instructions, theory, research-based facts, and workshop scenarios.-Main features: it helps young people hold workshops related to different competences, such as digital tools and resource suggestions, ready-to-use scenarios, timings, and other practical pointers.-Aims: it aims to enhance the competence of youth workers and educators by providing workshops for them and developing their 21st century employability skills.-Fields of use: IT and digital learning.


Toolkits in the field of electronics:(15)LittleBits [61]
-Main users: it is open to any type of user.-Main components: circuit boards and tiny magnets.-Main features: it provides an open-source library of discrete electronic components pre-assembled in a tiny circuit board.-Aims: it aims to help create complex structures and prototypes with very little engineering knowledge, similar to LEGO.-Fields of use: electronics.


Toolkits applicable in the field of social science:(16)SSH Training Discovery Toolkit [62]
-Main users: researchers, service providers, data stewards, and trainers.-Main components: educational materials and resources, e-learning modules, courses, workshops, slides, videos, games, reports, and computational notebooks.-Main features: it enables trainers to find a variety of materials they can reuse to develop and improve their own training activities.-Aims: it aims to provide a variety of training materials related to various topics, including research data management, open science, and didactics.-Fields of use: social sciences and humanities.
(17)Management Toolkit [63]
-Main users: corporate managers, business schools, consultants, and trainers.-Main components: resources and materials (seven core chapters, and three annexes).-Main features: it helps users in transition or developing economies; it also provides practical guide to define and implement programs such as strategic human resources management, management development, or other close disciplines.-Aims: it aims to support the design and implementation of management development and training programs.-Fields of use: management. 
(18)Research Leader’s Impact Toolkit [64]
-Main users: higher education institutions, senior research leaders, principal investigators, programmers, and project leaders.-Main components: a suite of research-based tools.-Main features: it helps users to develop or update a formal research impact strategy to, for example, build capacity, skills, and knowledge for research careers as well as to define strategies for leading, managing, and practicing impact.-Aims: it aims to provide a suite of research-based tools for higher education institutions to develop an embedded approach to research impact.-Fields of use: management. 
(19)Monitoring and evaluation methodology Toolkit [65]
-Main users: local government and public authorities.-Main components: training tools and procedures and course materials.-Main features: it helps public authorities to enhance the process of human resource management by providing needed concrete tools and procedures, it also promotes the sustainability of training systems and training programmes.-Aims: it aims to support the process of monitoring and evaluating the training programs provided for public employees.-Fields of use: (HR) management.
(20)Youth4Peace Training Toolkit [66]
-Main users: beginners and intermediate youth trainers and educators.-Main components: tips, explanations, and showcases.-Main features: it helps users to understand and apply the key concepts around peacebuilding (e.g., violence, conflict, peace, and transforming narrative); it also assists them to design and implement their activity.-Aims: it aims to support training on peacebuilding, conflict transformation, and creating peaceful narratives.-Fields of use: peacebuilding.


Toolkits applicable in the field of education & learning:(21)TESSA Inclusive Education Toolkit [67]
-Main users: educators, educational supervisors, teachers, instructors, and trainers.-Main components: it is a collection of resources and tools to refer to where certain challenges occur in the process of teaching practice supervision.-Main features: it helps users to explore and understand the meaning of ‘inclusive education’; it also provides a set of teacher training tools that can be adapted and used in different environments and contexts.-Aims: it aims to support the training of teachers in inclusive education and continuing professional development.-Fields of use: inclusive education.
(22)Gender-responsive education Toolkit [68]
-Main users: teachers, educators, and education professionals.-Main components: worksheets, instruction, assessment, case studies, methods for gender analysis, guidelines, frameworks, and online resources.-Main features: it guides the day-to-day practices of users in school management, teacher training, teaching and learning practices, and gender-responsive teaching materials. It serves as a resource to cultivate problem-solving, critical thinking, and innovative approaches in relation to gender mainstreaming in teacher training and learning practices.-Aims: it aims to support the improvement of the teaching methods and learning assessment techniques.-Fields of use: gender-responsive education.
(23)Teach for climate action: an advocacy Toolkit on climate change education for educators and their unions [69]
-Main users: labor market partners (employers and unions), policymakers, teachers/trainers, and training providers.-Main components: practices, frameworks, and recommendations.-Main features: it helps users to build up their baseline knowledge and skills; it also brings about a steppingstone for developing context- and user-specific plans for climate change education-focused advocacy.-Aims: it aims to build baseline knowledge and skills of educators and education unionists in climate change education.-Fields of use: climate change education.
(24)Toolkit for designing a comprehensive distance learning strategy [70]
-Main users: curriculum developers, policymakers, teachers/trainers, and training providers.-Main components: samples, online resources, frameworks, guidelines, self-reflection, and action planning.-Main features: it helps users to understand what distance learning is, how it works, and why it is important. It provides action points and tools that direct users in collecting and analyzing the needed data and making decisions for creating and developing their own distance-learning strategies.-Aims: it aims to support the designing of a comprehensive distance-learning strategy that covers an entire education sector or system.-Fields of use: distance learning.


Toolkits applicable in multiple disciplines and fields:(25)IHR Training Toolkit [71]
-Main users: IHR professionals (in the public health and security sectors).-Main components: it is a set of training resources, standard-quality material, and expert and peer support.-Main features: it assists users to prepare and train new generations of public health leaders and managers. It can provide a flexible interactive web-based tool for particular contexts and specific needs.-Aims: it aims to support the design and organizing of training modules on the IHR.-Fields of use: healthcare, food and agriculture, transport, travel, trade, education, and defense.
(26)roBlocks [72]
-Main users: children and young inventors.-Main components: sensors, logic, and actuator blocks.-Main features: as a computational construction kit, it encourages users to experiment and play with provided sensors blocks, actuator blocks, logic blocks, and utility blocks to understand the concepts related to feedback, distributed control, and kinematics.-Aims: It aims to aid in scaffolding children’s math, science, and control theory education.-Fields of use: mathematics, science education, technology, and engineering.
(27)World Café (dialogue) [73]
-Main users: activists and advocates (interested in organizing a dialogue-based film screening of American revolutionaries).-Main components: it (is a social technology that) consists of different guides, practices, lessons, processes, methods, or techniques (for engaging people in conversations that matter).-Main features: it provides a step-by-step guide for users to best practice and organize their own dialogue-based film screenings with success. It offers a sample agenda that users can adapt for their own events. It can be customized based on users’ objectives and needs.-Aims: it aims to support a dialogue with a group of activists working on a cross-section of issues.-Fields of use: it is open to any field and discipline.
(28)Ketso [74]
-Main users: universities, schools, businesses, public sector agencies, researchers, educators, and practitioners.-Main components: it is an array of information-gathering techniques (that utilize reusable-colored shapes to capture everyone’s ideas).-Main features: it enables users to think and work together better. It can be used in countless situations, settings, and ways across extension program areas (e.g., family and consumer sciences, natural resources, agriculture, aquaculture, and community development).-Aims: it aims to help people to collaborate, share their information, learn from each other, make decisions, and plan actions.-Fields of use: it is open to any field and discipline.
(29)After Action Review Toolkit [75]
-Main users: it is open to any type of team (who want to maximize learning from their work).-Main components: it comprises discussion techniques, reviews, frequent group process checks, notes, charts, and reports.-Main features: it helps users identify the strengths, weaknesses, and areas for improvement in their projects or events. It provides recommendations to overcome obstacles.-Aims: it aims to reveal what has been learned during and after a project to improve the organization’s performance, preparedness, response, and recovery.-Fields of use: it can be used in any project, program, activity, event, or task.
(30)Self-assessment Toolkit [76]
-Main users: course teams, senior managers faculties, and training institutions.-Main components: reports. -Main features: it helps to develop skill systems and support the development of relevant skills for all users.-Aims: it aims to improve the ability of training institutes to deliver high-quality learning and teaching across a wide range of curriculum programs.-Fields of use: it is open to any field and discipline.
(31)TalentLMS [77]
-Main users: employees, managers, customers, and partners of the business.-Main components: software’s course creation tools and training materials.-Main features: it helps users to design and develop various online training courses that fit their business needs. It is an intuitive, easy to learn, easy to use, and simple platform that offers built-in content creation.-Aims: as a learning management system, it aims to facilitate and expedite the creation and design of eLearning courses.-Fields of use: it is open to any field and discipline.
(32)Research Impact Toolkit [78]
-Main users: researchers.-Main components: research resources (e.g., videos, definitions, taxonomies, links, guidance, and case studies).-Main features: it enables users to identify who in the society can benefit from their research. It helps researchers to find out how much their work changes or benefits society, culture, economy, public policy or services, environment, health, and quality of life.-Aims: it aims to help researchers to plan, capture, communicate, and monitor the impact of their research.-Fields of use: it is open to any field and discipline.
(33)Digital Learning Toolkit [79]
-Main users: teachers, trainers, coaches, and learning designers.-Main Components: Teaching resources, teaching methods, and advice.-Main features: it helps users to convert their face-to-face education and training into digital and online learning formats. It also enables users to create engaging, active, and effective digital courses.-Aims: it aims to facilitate the design and delivery of successful online courses.-Fields of use: it is open to any field and discipline.
(34)Digital pedagogy Toolkit [80]
-Main users: professional developers of curriculum (e.g., teaching staff, librarians, and learning technologists).-Main components: recommendations, guidelines, and online resources.-Main features: it helps users to overcome barriers to using technology, and also it ensures users that the technology they use can meet the learning outcomes of the course, module, or programme of study.-Aims: it aims to support academic staff in planning, designing, and delivering the digital curriculum.-Fields of use: it is open to any field and discipline.
(35)Open education resources Toolkit [81]
-Main users: educators, teachers, training designers, and providers.-Main components: guidelines, checklists, and online resources (accompanied by a checklist of questions).-Main features: it helps users to use, create, and publish education resources. It also promotes the quality of learning and teaching experiences, research, open science, and open access publications.-Aims: it aims to guide the process of course design and materials development.-Fields of use: it is open to any field and discipline.
(36)Postsecondary education and training preparation Toolkit [82]
-Main users: students with intellectual disabilities, family members, service providers, and educators.-Main components: guidelines, online resources, and research.-Main features: it helps young adults with disabilities to benefit from an array of opportunities, services, and programs, aiming to gain success in postsecondary education and training.-Aims: it aims to support students with disabilities to gain skills for future employment or better employment, to develop important life skills, and to engage in learning and living with other young adults.-Fields of use: it is open to any field and discipline.


The partial analysis presented above can increase our understanding of the characteristics and functions of the Toolkits identified. The fact is that we tried to present the most important information about the Toolkits. However, we neither could find the strengths and weaknesses of all the identified Toolkits, nor could we compare the Toolkits functionalities and capabilities based on their common strengths and weaknesses. Therefore, in order to facilitate the process of decision-making for choosing the right Toolkit for a particular purpose, Table 1 provides a comparison of the Toolkits by taking into account the three major factors (supporting students to develop their skills, supporting teachers in training or course development, and supporting multiple training scenarios) that are considered in the services and solutions we proposed. The Toolkits that show signs of connection with each considered major factor are marked with (X) in Table 1.

The Toolkits listed above each have the potential to be used exclusively or in combination with other Toolkits to make a specified package of training Toolkits for a specific case and group of users. In the implementation of a package and grouping of multiple training Toolkits, some important factors need to be taken into account, including the purpose and requirements of the training and learning program, as well as the financial, cultural, and political conditions of the learning ecosystem (e.g., Smart Lab), to name a few. 

### 3.3. Data Analysis (Partially S3)

The analysis of the identified Toolkits stands on the range of applications and functional capabilities that they can represent. As shown in Table 2, we proposed and considered 7 groups of application and functional capability for the identified Toolkits. These groups are proposed based on (a) the main components of the Toolkits, (b) the goal of using the Toolkits, and (c) the functionality of the Toolkits. If the number of Toolkits shown in Table 2 increases, the number of classification groups might increase too, and vice versa. This classification helps us better organize and present the Toolkits. Having proposed the group classification through a theoretical and conceptual assessment, the Toolkits that show signs of connection with each considered group of applications and functions are marked with (X) in Table 2 (partially covering the third stage of our methodology).

Taking Table 2 into account, the analysis shows that the majority of Toolkits (72.22%) are designed to be applied for the purpose of ‘didactics learning’, followed by application as ‘technology’ and ‘consultation’ with 30.55%. The lowest degree (8.33%) of applications is given to Toolkits that are made to be used for/as ‘presentations & documents’ and ‘engagement & discourse’. This analysis gives us an indication of the types, numbers, and percentages of the group of applications and functions addressed by the identified training Toolkits. 

### 3.4. Proposed Model (S4) 

Our proposed model—which is demonstrated in Figure 3—simply conceptualizes and represents a typical Smart Lab and a general package of training Toolkits that can potentially support users (addressing the fourth stage of our methodology). Taking into account the 7 groups of Toolkits proposed in Table 2, we first identified a number of instances of the related training Toolkits and then classified them under the 7 groups, according to their application and functional capabilities. It is important to note that the number of groups and their associated Toolkits and instances should be adjusted according to the goals and requirements of the concrete case. As an example, in reliance on our literature review, we added one more group (group 8) to the list, aiming to make the model more comprehensive, although this group (8) was not supported by the Toolkits listed in Table 2.

In order to have a clear picture of the Toolkits addressed in Figure 3, some examples and instances of the identified Toolkits used for discovery (group 8) are presented in Figure 4. It is believed that these Toolkits not only used for discovery and finding things out but are also helpful for conducting self-learning. 

It should be emphasized that our conceptual model is proposed to guide the educational institutes and faculty members who want to design and build up their own package of training Toolkits for supporting specific discipline(s)/module courses and developing students’ skills, which are not often included in traditional education systems.

## 4. Instantiation and Specification Cycle

This section presents the instantiations we used to test our proposed conceptual model, addressing the 1st and 2nd stages of the instantiation and specification cycle and the 5th and 6th stages of our proposed workflow methodology. Section 4.1 and Section 4.2 present two examples of instantiations we provided. 

### 4.1. Instantiation for the Toolkit—B-Health Box (S5)

Nowadays, many people around the world suffer from different types of physiological and posture-related problems caused not only by the incorrect execution of their work but also by not receiving the needed advice and/or corrective measures. To take a step in solving this problem, the B-Health Box Toolkit was designed (covering the fifth stage of our methodology), which relies on posture data collection and uses smart sensors to monitor students’ physical activity and their posture. The idea is to trigger real-time corrective notifications to students (users), intending to reduce the risk of physical problems and injury that might be imposed on them as a result of not taking correct physical positions. The notification might be coupled with some advice and useful information (e.g., to correct their posture or relax their shoulders and back muscles) that indicates the causes affecting their physical condition, aiming to provide preventive measures within their daily environment. 

In line with the healthcare objective mentioned above, the Toolkit also aims to help students learn how to harvest data from IoT devices and how to simulate human intelligence processes by machines, especially computer systems. The knowledge and skills gained in this way enable students to program machine learning algorithms that can automatically detect whether or not the student’s posture is correct. 

Figure 5 demonstrates the main materials used in this Toolkit. The image on the left side of the Figure shows the B-Health Box; the image in the center shows the Physiosense posture sensor: the sensor with the case (upper image) and the Printed Circuit Board (PCB)—(below image); and the image on the right side shows the T-shirt (to be used by the student). 

Figure 6 demonstrates the package of training Toolkits provided by B-Health Box and clarifies the way in which this Toolkit can be used to help students develop their skills.

To give an example, the B-Health Box can ‘provide’ students with at least 4 types of technology Toolkits (through group 1) namely, training/research tools, applications, sensors, and computational notebooks. Each of these technology Toolkits can in turn support students in developing their skills in the fields of IoT and AI.

It should be added that this model, at this stage of development, faces two main limitations. The first limitation is with the number of applications of the model. That is, the model was only applied to our proposed services and solutions, although it needs to be validated by further applications. Second, the model was limited to the contexts of AI, IoT, OS, and interoperability. However, it should be applied to and tested in other contexts and fields of study.

#### Using B-Health Box Toolkit (S6) 

To make the connection between the B-Health Box and the user, one of the Phisiosense posture sensors should be placed in the pocket (designed for this purpose) of the T-shirt (used by the student), below the neck and between the shoulders, and another one in his/her lower back. The Physiosense posture sensors were developed to be connected via Bluetooth protocol. With this protocol, the students can use the sensors on their personal cell phones and also benefit from the Citizen Hub application. This application allows the students to receive feedback from the Physiosense posture sensors, indicating—on their cellphones every time—that they have an incorrect posture. To detect the students’ posture, we use an IMU (Inertial Measurement Unit) sensor. More specifically, we apply the accelerometer sensor to use the three axes for detecting posture variations. To identify the incorrect posture, we use two Physiosense posture sensors that are placed in the pockets, as mentioned above. In this way, we can check the inclination of the back more accurately and compare the inclination between the two sensors. With this approach, it is possible to verify the inclination of the posture by using two of the three considered axes. The two Physiosense sensors are used to make a correlation between the two axes, and in this way, it is possible to verify the vertical inclination of the back and give warnings whenever one of the students leans forward (representing an incorrect posture). At the same time, the horizontal axis of the sensors starts checking to identify strange positions that the students might put on their backs, for example, when the back is not aligned with the body (see Figure 7). After collecting the data from the Physiosense posture sensors through the B-Health Box, it is necessary to process that data. In this way, the students will have to develop an AI algorithm per group of students that identifies when a student has an incorrect posture. Furthermore, it sends him/her warning notifications whenever it occurs. The related AI algorithm will be developed in Python. At the end, the students will be able to use the B-Health Box to execute the components, so that their posture will appear in real time on a monitor as shown in Figure 7.

The B-Health Box contains a Raspberry Pi 4, and the students will learn to work in Linux. That is, they have to develop Python code to communicate with the Physiosense posture sensors through Bluetooth. Thus, they need to learn the Bluetooth communication protocol to understand how it works and how to program it. The developed code must connect to the sensors and start receiving data in real time that will be saved in a csv file (comma-separated values—https://en.wikipedia.org/wiki/Comma-separated_values) (access date: 20 February 2023). Given that, this Toolkit helps students develop code components for the B-Health Box capable of communicating with the Physiosense posture sensors through the Bluetooth communication protocol and of receiving in real time the data from the sensors that will be later used by an algorithm developed by students. Therefore, this Toolkit sends a visual warning to the students when their postures are incorrect, and in this way, the students can proceed to the placement of the correct posture. This section covers the sixth stage of our methodology.

The Toolkit is a self-contained problem box with known parameters. It provides a concise and clear development platform to explore real-world conditions for Bluetooth, IoT, and Data Collection. The Toolkit is a ready-to-use solution, which should free the teacher to concentrate on the tasks to be presented to the students.

### 4.2. Instantiation for Discipline and Module Course (S5)

To demonstrate the usefulness and effectiveness of our proposed workflow methodology and conceptual model for developing disciplines and module courses, we took the opportunity to apply them in the LLSF project (project education 4.0: living labs for the students of the future) [83]. This project addresses the services and skills necessary for the digital industry of the future and will develop an international network of interconnected labs that offer flexible options for digital study to Master’s and Ph.D. students. The project aims to pilot a program for Living Labs showcasing a study on the creation of education and innovation value through the introduction of digitally interconnected Smart Labs, toward introducing smart laboratories, where the learning experience is centered around DATA as well. 

NOVA University of Lisbon [84], as a partner involved in this project, has adapted and applied the proposed workflow methodology and conceptual model for upskilling a number of students and developing a discipline delivered by the Centre of Technology and Systems (CTS) [85]. CTS is located in the Faculty of Science and Technology (FCT) of NOVA University of Lisbon and its mission is to carry out fundamental and applied research, advanced training, dissemination of knowledge, and encouragement of technology transfer in the main areas of Electrical and Computer Engineering (EEC). 

The considered discipline for development is Architecture for Integration of Systems (AIS), which aims to enhance students’ understanding of the whys and hows of today’s common problems at the application and business levels, which are directly or indirectly caused by interoperability difficulties. The discipline offers a module course that helps students develop their skills and capabilities related to IoT and AI, as well as their research and technical solutions for interoperability problems. Accordingly, the module course provides several related syllabuses, including an introduction to the course, an interoperability module, an IoT module, an Operation System (OS) module, an AI module, and a practical module. This module course, in addition to providing theoretical material (through the syllabuses), provides some opportunities for students develop their skills practically in Linux and Python by means of connection to a specified Smart Lab and working with the B-Health Box and its sensors and controllers, which were mentioned earlier. 

It should be mentioned that a total of 30 master’s students who took the AIS discipline were given access to the Smart Lab and allowed to test the B-Health Box. Having evaluated the feedback of this group of students, another group of master’s students will then test the B-Health Box, aiming to improve any weaknesses that might be found. NOVA University of Lisbon will then ask the students of the other 4 universities that are partners in the project to test the B-Health Box. The direct beneficiaries who test the B-Health Box are around 200 MSc students and 6 PhD students. The B-Health Box will then be tested by other types of users, such as teachers and stakeholders of the project.

Since one of the main objectives of the work is to help teachers create training programs incorporating Smart Labs and its Toolkits, the authors specified lateral stages between the main stages of the instantiation and specification cycles. Thus, Figure 8 illustrates the main and lateral stages (of the instantiation and specification cycle) that are associated with the instantiation of the disciple and module course to be taken into account in such an objective. 

As illustrated in Figure 8, there are two groups of lateral stages for the teachers. The first group (S5a, S5b, and S5c) is added after S5 (instantiation stage), and the second group (S7a, S7b, S7c, and S7d) is added after S7 (evaluation stage). Figure 9 demonstrates the processes that the teachers should follow to design and develop their discipline and module courses, as well as to use the service provided through this instantiation.

As shown in Figure 9, the process starts with the first lateral stage (S5a). At this stage, the teacher(s) take into account the results of evaluating the students’ current skills to identify which skills need development. At stage S5b, the teacher(s) try to establish the objectives of their discipline and/or module course that they want to deliver and/or develop. This stage allows teachers to plan, organize, and control the sequence, time duration, and resources required to ensure the success of the discipline or module course. At stage S5c, the teachers proceed with the configuration of the package and selection of the associated Toolkits that can support the delivery and/or development of their discipline/module course. At Stage 6, which is one of the main stages of the cycle, the teachers use the provided services and solutions (Smart Lab and the associated package of Toolkits) for delivering and/or developing their discipline/module course. At stages S7a, S7b, and S7c, the teacher(s) evaluate the effectiveness and efficiency of the provided discipline/module course, Smart Lab, and the associated package of Toolkits. If the teachers are satisfied with the provided discipline/module course, Smart Lab, and the associated package of Toolkits, they can then keep using them. Otherwise, the university and services provider(s) should be informed to take the needed actions. At stage S7d, the teacher(s) make another evaluation by comparing the results of the students’ skill levels before and after using the service. If the results of this evaluation show that the students’ skills were developed insufficiently, the teacher(s) should then proceed to stage 8 and alter or modify their package of Toolkits. If the results of this evaluation indicate that the components and features of the Smart Lab should be improved in order to better classify the Toolkits, Furthermore, at Stage 9, the teacher(s) need to improve the Smart Lab accordingly. 

As mentioned previously, the main contribution of NOVA University of Lisbon to the project consists of the designed AIS, the specified Smart Lab, and the B-Health Box. The other partners and universities that have contributed to the project will also propose and introduce their own services and solutions including, (a) the disciplines and modules that use a range of teaching and learning strategies and materials with a view to foster a deeper knowledge of subject areas and creating transferable 21st-century skills, (b) the Smart Labs that can support the implementation of digital tools to be then used by students for acquiring the necessary skills, and (c) the training Toolkits that can make synergy between foundational (hard) skills and (soft) 21st-century skills. 

Having analyzed the features and capabilities of the proposed and provided services and solutions (by all partners of the project), the consortium will then select, adapt, and accommodate a set of commonly used services (by almost all the partners and universities) in a single package of supporting services to be used by all involved participants, aiming to unite the many facets of training service provided as well as to create harmony and synergy between the provided services, service providers, and users. The users (teachers and students), thus, can connect, collaborate, and share the best practices with each other through a networked supportive training service and reap the long-term benefits of the created operational synergy toward the betterment of their performance and outputs. 

## 5. Conclusions

To stay competitive in the global market and keep up with developments in technology, it is essential for businesses to invest in highly skilled workers. In a trend that seems long overdue, businesses are increasingly turning inward to bridge the gap between the skills they need workers to have and the skills they are actually graduating from school with. As skilled workers become more and more in demand, many educational institutes around the world have made an attempt to train students (workers of the future) with the required skills. The fact is that the true ‘21st-century learning system’ needs to change the way in which students are trained and should prepare them for modern careers and the jobs of the future. From this perspective, the proposed solutions not only should encourage students to learn pure content and more in need of skills for mapping out and building successful career paths but also should simultaneously empower teachers with effective tools and resources to train students with the core technical skills needed. Given that, the contribution of this study is proposing a workflow methodology that can guide and support educational institutions (particularly higher education institutions) in the implementation of training Toolkits in Smart Labs. As such, we proposed a conceptual model, representing a generic package of training Toolkits that contains eight groups of Toolkits and their related examples and instances. The package is adaptable for the development of both (a) different disciplines/module courses and (b) students’ practical and professional skills. The utility and effectiveness of our proposed services and solutions are tested through the specified discipline and the designed B-Health Box. The primary results gained from the application and observational evaluation of our proposed services and solutions in the used case are satisfactory, indicating that they can be potentially used in other case studies toward their validation. 

### Future Work

It should be mentioned that our proposed services and solutions are still in progress. At the time of closing this paper, we are at the evaluation stage (S7). The primary results gained from our observational evaluation and recording of the feelings of teachers and students about using the provided services show that our proposed solutions are appealing to and helpful for the users. In future work, we will publish the results of our further evaluation, presenting the extent to which the provided services and solutions are found useful, efficient, and effective by the users of the services. The process of evaluation will be conducted by means of a web portal that we are now trying to design (at the stage of closing this work). The evaluation could be performed, for example, by providing online questionnaires that should be filled out by the users. The results of evaluations will then be stored on the portal for consideration and further development. 

## Figures and Tables

**Figure 1 sensors-23-02626-f001:**
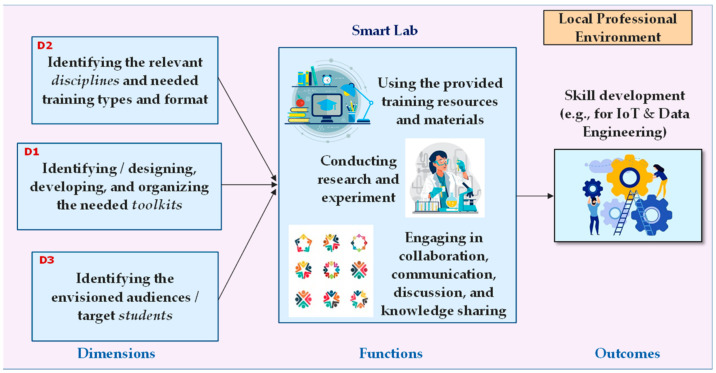
A matrix model, addressing the main dimensions and functions for developing the industrial and technical skills of students.

**Figure 2 sensors-23-02626-f002:**
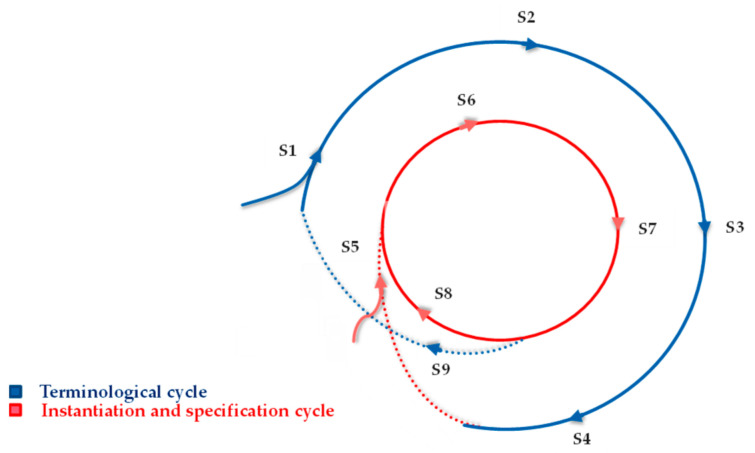
Proposed workflow methodology for training Toolkit implementation in Smart Labs.

**Figure 3 sensors-23-02626-f003:**
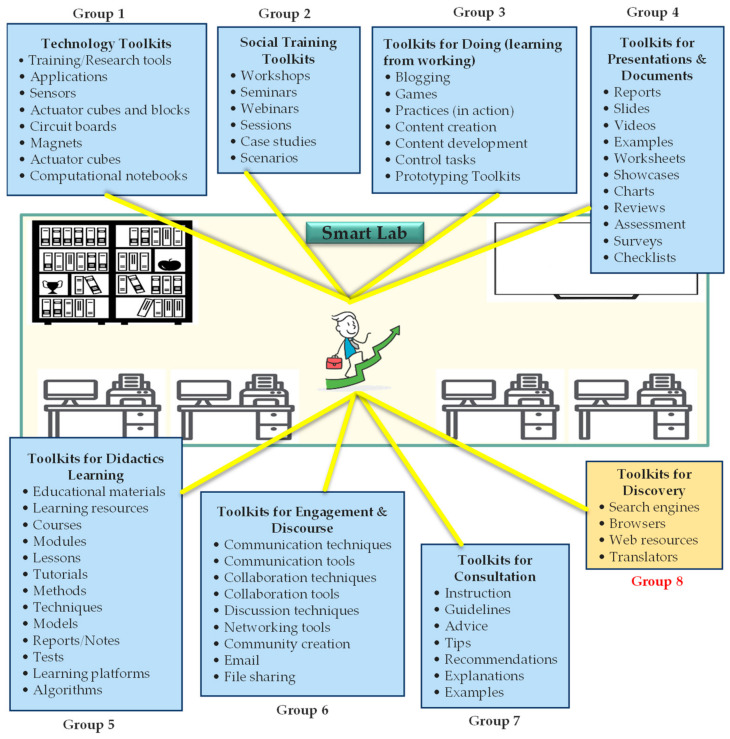
Proposed conceptual model for creating a package of training Toolkits in the Smart Labs.

**Figure 4 sensors-23-02626-f004:**
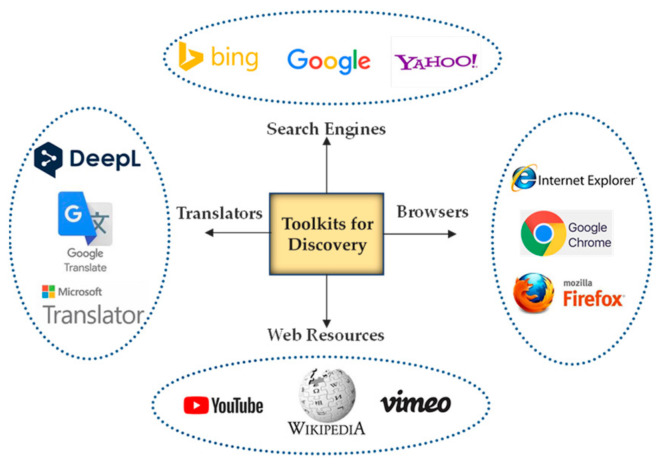
Sample of Toolkits for discovery.

**Figure 5 sensors-23-02626-f005:**
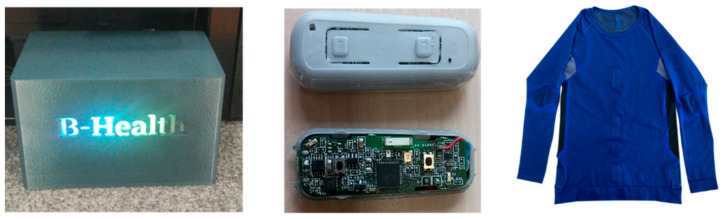
B-Health Box, Physiosense Posture Sensors, and T-shirt.

**Figure 6 sensors-23-02626-f006:**
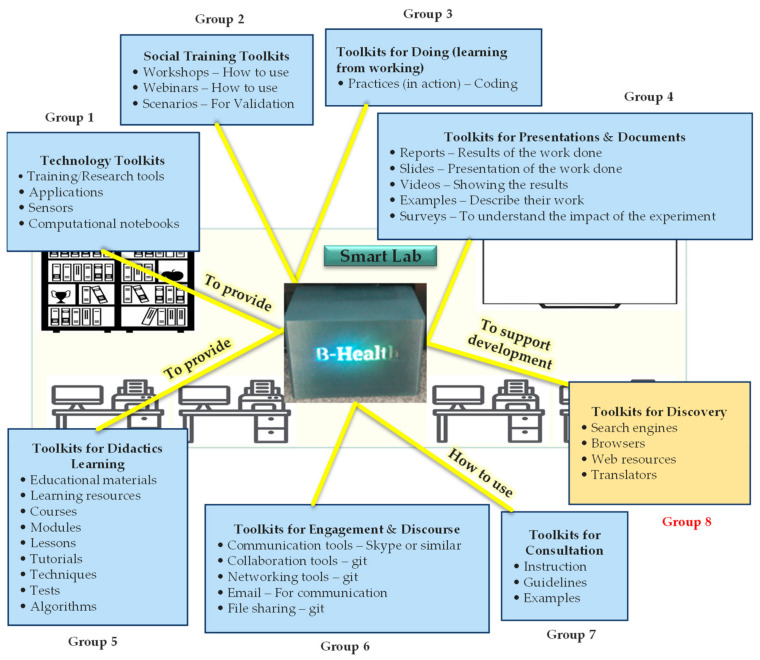
Package of training Toolkits provided by B-Health Box implementation.

**Figure 7 sensors-23-02626-f007:**
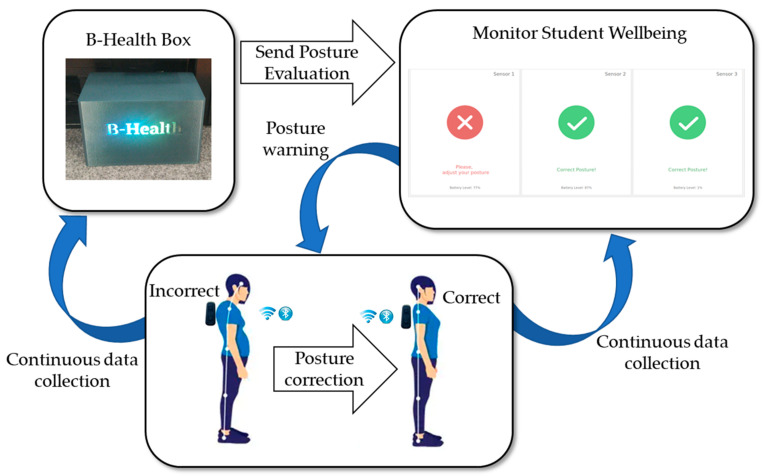
B-Health Box implementation.

**Figure 8 sensors-23-02626-f008:**
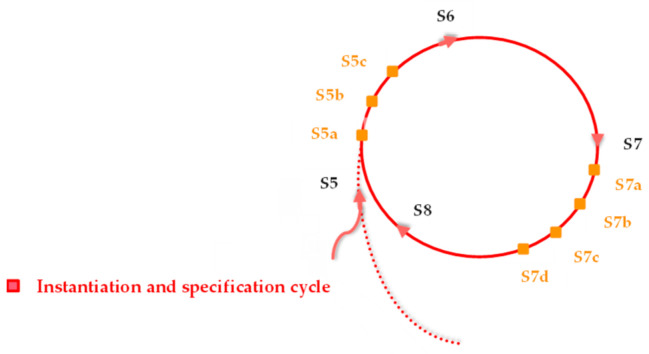
Sample alteration (increase) in the number of stages for teacher users.

**Figure 9 sensors-23-02626-f009:**
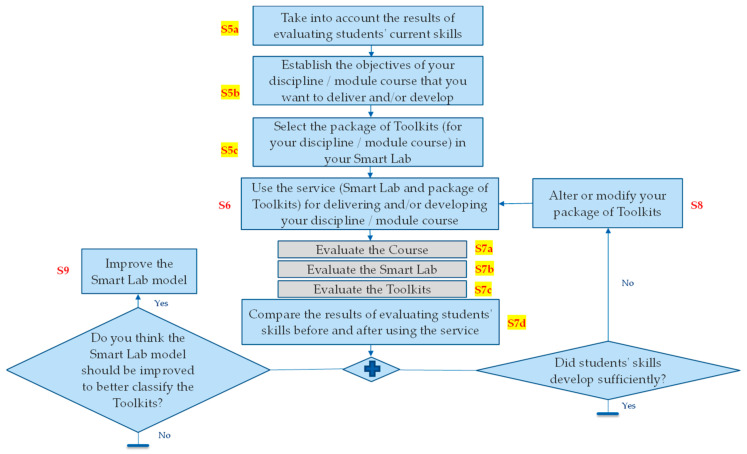
Proposed process for designing and developing the disciplines and module courses.

**Table 1 sensors-23-02626-t001:** Analyzing the identified Toolkits based on the types of support they provide.

Toolkits	Support Students to Develop Their Skill	Support Teachers in Training/Course Development	Support Multiple Training Scenarios
1. Unity ML-Agents Toolkit	X	X	X
2. OpenAI Gym toolkit	X	X	X
3. TensorFlow toolkit		X	X
4. DeepMind Control Suite		X	X
5. LEAF		X	X
6. Training toolkit		X	
7. Laboratory Quality Management System Training Toolkit		X	
8. Training Toolkit		X	X
9. STEM educational toolkit	X		X
10. Real-time Distributed Toolkit	X		X
11. ConnectUs	X		X
12. MakeBlock AI & IoT Education Toolkit Add-on Pack	X		X
13. Talkoo Toolkits	X	X	X
14. Training Toolkit	X		X
15. LittleBits	X	X	X
16. SSH Training Discovery Toolkit		X	X
17. Management toolkit		X	X
18. Research Leader’s Impact Toolkit		X	X
19. Monitoring and evaluation methodology Toolkit		X	X
20. Youth4Peace Training Toolkit	X	X	X
21. TESSA Inclusive Education Toolkit		X	X
22. Gender-responsive education Toolkit		X	X
23. Teach for climate action: an advocacy toolkit on climate change education for educators and their unions		X	X
24. Toolkit for designing a comprehensive distance learning strategy		X	X
25. IHR Training toolkit		X	
26. roBlocks	X		
27. World Café (dialogue)		X	X
28. Ketso		X	X
29. After Action Review toolkit		X	X
30. Self-assessment toolkit		X	X
31. TalentLMS		X	X
32. Research Impact Toolkit		X	X
33. Digital Learning Toolkit		X	X
34. Digital pedagogy toolkit		X	X
35. Open education resources toolkit		X	X
36. Postsecondary education and training preparation toolkit	X		X

**Table 2 sensors-23-02626-t002:** Analyzing the identified Toolkits based on their potential applications and functional capabilities.

Toolkits	Technologies	Doing (Learning from Working)	Presentations & Documents	Didactics Learning	Consultation	Social Training	Engagement & Discourse
1. Unity ML-Agents Toolkit				X			
2. OpenAI Gym toolkit		X					
3. TensorFlow toolkit	X			X			
4. DeepMind Control Suite		X		X			
5. LEAF				X			
6. Training toolkit				X			
7. Laboratory Quality Management System Training Toolkit				X	X		
8. Training Toolkit				X			
9. STEM educational toolkit				X			
10. Real-time Distributed Toolkit	X			X			
11. ConnectUs	X						
12. MakeBlock AI & IoT Education Toolkit Add-on Pack	X			X			
13. Talkoo Toolkits		X		X			
14. Training Toolkit					X	X	
15. LittleBits	X						
16. SSH Training Discovery Toolkit	X	X	X	X		X	
17. Management toolkit				X			
18. Research Leader’s Impact Toolkit	X						
19. Monitoring and evaluation methodology Toolkit	X			X			
20. Youth4Peace Training Toolkit					X		
21. TESSA Inclusive Education Toolkit	X			X			
22. Gender-responsive education Toolkit				X	X	X	
23. Teach for climate action: an advocacy toolkit on climate change education for educators and their unions		X			X		
24. Toolkit for designing a comprehensive distance learning strategy				X	X		
25. IHR Training toolkit				X			
26. roBlocks	X						
27. World Café (dialogue)		X		X	X		X
28. Ketso							X
29. After Action Review toolkit			X	X			X
30. Self-assessment toolkit				X			
31. TalentLMS	X			X			
32. Research Impact Toolkit			X		X	X	
33. Digital Learning Toolkit				X	X		
34. Digital pedagogy toolkit				X	X		
35. Open education resources toolkit				X	X		
36. Postsecondary education and training preparation toolkit				X	X		
**Numbers**	11	6	3	26	11	4	3

## Data Availability

Not applicable.

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
