# Peer review of "A Methodology for Training Toolkits Implementation in Smart Labs"

_sensors, 2023, doi:10.3390/s23052626_

Round 1

Reviewer 1 Report

In section 3.2.2.1. Identified Training Toolkits - it could be useful to have a synstesis on matrix in order to have a fast modality to compare the difference among the analised tools. Please add the weekness and the strengths for each of them. otherwise, the analysis you have done risks being just a simple one list. Something similar to what have been done in section 3.3

It is not clear what the limitations of the tools in general are and what you have added instead compared to the state of the Art.

Regarding the conclusions: on how many students have you tested the Toolkit? Could you add some quantitative considerations to effectively bring out the usefulness of the toolkit and the model you developed?

Author Response

Dear Reviewer,

Please find the annex, our response to your comments.

Reviewer 2 Report

Thanks a lot for giving me a chance to review this paper. The is well written and presents a very interesting idea. However, I suggest the following minor corrections.

1. I see some spelling mistakes. for example (line 209). Please check the whole paper for such errors.

2. Please add the limitations of the proposed model. It should be noted that such limitations are very important and eliminate what was out of the scope.

2. Future work is also not there. Please add a separate section after conclusion.

Author Response

(The authors gave the same response as above.)
